# Jeong: A Practical Theology of Postcolonial Interfaith Relations

**Sue Kim Park**

Practical Theology, Columbia Theological Seminary, Decatur, GA 30030, USA; parkS@ctsnet.edu

**Abstract:** This article examines Korean American Christians' involvement in interfaith relations from a practical theology perspective. The author begins the research with the broad question, "What is going on with Korean American Christians in interfaith engagement?" and interrogates more specifically the methods through which they participate in it. Gathering results from ethnographic research, the author claims that Korean American Christians build interfaith relationships through jeong, a collective sentiment many Koreans share. Jeong is an emotional bond that develops and matures over time in interpersonal relationships. As for interfaith engagements, Korean American Christians cultivate organic, messy, affectionate, and sticky relationships, letting jeong seep into their lives across religious, faith, and non-faith lines. The praxis of jeong is analyzed in three categories: (1) love and affection, (2) liberating and healing power, and (3) stickiness and vulnerability.

**Keywords:** interfaith; interreligious engagement; han; jeong; Korean American Christianity; practical theology; postcolonial

## 1. Introduction

Interfaith dialogue, as practice and the subject of scholarly inquiry, has been growing steadily worldwide. Conflict in the name of religion and oppression across religious lines are two obvious reasons compelling religious leaders to collaborate to seek peace and justice together and seek interfaith dialogue as a form of spiritual practice or to strengthen one's own faith community. In this article, I will analyze interfaith engagement from Korean American Christian (KAC) perspectives. Practical theologians are taught and trained to ask questions to investigate and discover how theologies are practiced in the lives of the faith communities. As a practical theologian, I ask, "What is going on with interfaith engagement in my KAC faith community?"

My commitment to interfaith dialogue highlights the apparent absence of interfaith engagement in my own community. I have identified three related factors that can be attributed to KACs' lack of participation and interest in interfaith engagement. The first factor is KACs' conservative evangelical theological leanings and tendencies in general. Secondly, the existing interfaith methods and gatherings do not fit into the ways that KACs practice their faith. The third factor has much to do with the exclusivity of the interfaith methods and gatherings. KACs do not readily have access to interfaith gatherings unless they actively seek out these opportunities.

In this article, as a practical theologian living in a postcolonial space, I will provide a holistic analysis of the KAC context and the issues that may hinder or discourage KACs from being open to interfaith engagements. While my research shows that KACs do not have a presence in interfaith gatherings, I make a counterclaim that KACs are in fact actively engaging in organic, informal, interfaith relations. I also question how we categorize interfaith dialogue/engagement/interaction. I propose to analyze KACs' implicit engagement in interfaith relations as a praxis of jeong. I argue that KACs are engaged in interfaith relations without realizing it; this implicit engagement must be made explicit

both to KACs themselves and to the interfaith scholarly and practitioner world. I will begin with definitions and distinctions.

## 2. Context

I want to draw some parameters for the ways in which I use the term "context" before I address KACs' involvement and engagement in interfaith relations. The complexity around the term must be understood and inspected. This article comes from the particular context of the Korean American immigrant situation. This situation is not and should not be treated as a background story with which one familiarizes oneself to interact with KACs in interfaith engagement, but rather as an interconnected wholeness that is a real, tangible, and concrete reality that KACs live into and out of in our time. Therefore, we must understand the interconnected reality that weaves together Korean history, Korean Christian history, Western Christian histories, Western colonization, immigration history, diaspora implications, and power and privilege. It is an active part of KAC life and is still evolving as it dynamically interacts with the changing world.

Using Courtney Goto's analysis, I will delineate my use of the term "context" in this article. Goto clearly expresses her dissatisfaction with the ambiguous and general use of context to refer to a wide range of specific circumstances. To build a case for a more specified usage of context, she offers a practical theologian's critique on the objectivism that some researchers claim in their studies and uses Thomas Kuhn's epistemology to argue against objectivism in research. Kuhn's understanding of knowledge is that "knowledge and knowledge production are deeply shaped by the needs, loyalties, and social dynamics of the investigators themselves."[1] The relationship between context and researcher is more complex than one may imagine and, thus, claiming objectivism in one's research "erases not only the knower, her social location, and her community but also the setting or the particulars of the object of study so that the investigator discovers the universal truth that lies behind them."[2] The researcher must be highly aware of what context means, how it is used, and how it interacts with the researcher. Without such awareness, the researcher may commit violence of erasure on many communities and research partners.

Goto gives four uses of context and three approaches to context to highlight its complexity and nuances and the necessity for researchers to clarify their uses and approaches to it. The four uses of context are (1) "social milieu," which refers to "the multiple circumstances (historical, demographic, religious, cultural, economic, legal, political, and aesthetic) in which the subject of research is situated"; (2) "a framing device in common speech"; (3) "a *background story* (including a series of events, characters, and commitments linked by a beginning, middle, and end)"; (4) "a *locus of concern*" which refers to the fact that "*various circumstances are presumed to be embedded in the particular/place time under investigation, making the locus of concern what it is*."[3] Definition four is particularly relevant for this article. When I refer to the KAC context, I am referring to the wholeness of the experiences of being immigrants, a minority, a certain type of Christianity with unique practices and history, and a certain class. This context cannot be separated into compartments or simply fade into the "background story" from which my claims arise.

One last point on context for this article is that I want to avoid analyzing the KAC context in terms of Western or global culture. The Korean American Christian context, as the name suggests, has components of Western, global, migration, and postcolonial cultures. Therefore, the lens through which Korean American Christianity is analyzed is postcolonial because Western worldviews and culture should not be centered or assumed to dominate the KAC context.[4]

---

1　(Goto 2018, p. 37).
2　Goto, *Practical Theology*, 27.
3　Goto, *Practical Theology*, 87–88, italics original.
4　Goto, *Practical Theology*, 103.

### 3. The Korean American Christian Context and Hindrances to Approaching the "Interfaith Table"

In an ethnographic study, I interviewed second-generation Korean Americans who grew up in the US to investigate KACs' involvement in interfaith dialogue.[5] These interviews were conducted both in person and via video conferencing as they were located in various parts of the US. The age range of the participants was from twenty years old to thirty-five years old. Many were actively involved in church community life and some held lay leadership positions while none of them did received any formal theological training. From the results of the interviews and analyzing artifacts including church bulletins, songs they sing, and sermon titles, I identified three factors that hinder KACs from actively seeking out interfaith interaction opportunities: conservative theological tendencies, interfaith engagement methods that do not fit KACs, and lack of access to interfaith events.

A major obstacle that discourages KACs is Korean Christian theological tendencies. Korean Christian churches for the most part lean toward conservative evangelicalism. While mainline Protestant denominations avoid using the term "evangelicalism" to describe their theological beliefs and doctrines, Korean and Korean American Christian churches embrace this fully. The basic definition for being an evangelical, even though it is a broad term that encompasses many aspects of the American society, is

> A Protestant who has made a definite personal decision to make the person of Jesus Christ as revealed in the Gospels her Savior and her Lord, and will go door to door—or pursue some other form of deliberate witness and evangelism—to persuade you and me and anyone else to make the same decision.[6]

The distinct characteristic of an evangelical Christian is the emphasis on personal conversion, which will express itself outwardly in an act of evangelism. Though there are far more socio-political implications that are associated with the label of conservative evangelical, personal conversion and evangelism lay sufficient foundation to understand Korean and Korean American Christian theology. With this understanding of a personal decision to follow Christ as the central message, KACs are taught to keep Christianity both pure and personal.

Being and remaining pure to church matters connotes an uncontaminated and original form of Christianity, following the examples of the early church. Western missionaries' evangelistic and social assistance efforts such as building hospitals and schools inevitably introduced the Western colonial narrative as a part of the (Western) Christian narrative. There are two related postcolonial assumptions: first, Western Christianity is superior, and second, there is little room for Korean culture in Christian theology. Other than the stories found in the Bible, the closest "original" form of Christianity they can trace back to is the one the Western missionaries brought. Deeply rooted in that brand was Western hegemonic authority, which usurped the subjects' autonomy and agency to claim their own religious identity.[7] In other words, Koreans received Western Christianity under the assumption that the Western form of Christianity was *the* Christianity, with full authority that did not allow room for their own interpretation. An example of this is the way Korean Protestant churches treat ancestor veneration, *chesa*. Protestant missionaries had a clear understanding and agreement before they even landed in Korea that ancestor worship was contrary to Christian teachings.[8] Many early converts suffered when they refused to bow down to the chesa tablets, but they were convinced that participating in chesa was committing idolatry. Even in the present day, Korean Protestant churches prohibit chesa even as

---

5　Koreans who immigrated to the United States are referred to as "first-generation Korean Americans." Those who were born in the United States are referred to as "second-generation Korean Americans." Those who immigrated with their parents when they were children are often referred to as "1.5-generation Korean Americans." In this article, I will be referring to all three categories of immigrants.

6　(Shah and Forster 2016, p. 142).

7　(Bhabha 1994, p. 86).

8　(Kim 1988, p. 27).

cultural expressions but have Christianized the practice as memorial services to remember and honor the deceased. Western hegemony in the Christian narrative was only reinforced with the Korean War and Western cultural, political, and military presence in Korea.

The second assumption stems from the first, as accepting Western Christianity and its theology as superior without much resistance led to the underdevelopment of Korean indigenous theology. Korean Christian theologians and clergy did not find creative ways to indigenize Christianity with Korean culture and spirituality. In the intersection of new Western Christianity and Korean culture is the space in which Bhabha's use of mimicry and mockery as a subversive tool would thrive; in this very interstitial space is where the colonial subject might creatively use the colonizer's tactics against them.[9] Bhabha shows how a group of villagers outside Delhi creatively resisted the dominant narrative and refused to receive Western Christianity without alterations. When presented with the Gospel, baptism, and eucharist, the villagers said, "We are willing to be baptized, but we will never take the Sacrament. To all the other customs of Christians we are willing to conform, but not to the Sacrament, because the Europeans eat cow's flesh, and this will never do for us"[10] These villagers did not seek to keep purity in Christianity, but they contaminated the colonial message with their indigenous culture.[11] From this vignette, Bhabha asserts that the natives were using "the power of hybridity to resist" and decenter the colonial narratives.[12] Korean Christians did not use the power of hybridity in decentering and contaminating the dominant narrative.

The second obstacle is that most interfaith engagement methods utilized in the West do not fit KACs' theological tendencies. When public engagement and practice of faith is discouraged and rather muted, the public nature of interfaith engagement methods do not appeal to KACs, and even if they did participate in these interfaith gatherings, they would not feel authentic. It would be remiss to not mention the need to shift the theological leanings and church teachings to include more engagement outside the church and bring to their attention that pure Christianity is a false notion. However, in this article, I am examining the ways in which KAC have and have not been involved in interfaith engagements, not calling on the KAC to educate differently.

The third obstacle is that while these methods work beautifully for individuals and communities who have full access to information about these gatherings and a pathway to them, such as invitations and social capital to attend formal meetings, KACs do not readily have access to them. Access is an issue for KACs mainly because they belong to immigrant faith communities who operate under a different set of cultures and languages. Even though KACs live with Christian privilege, they are ethnically a minority in the United States, and they face the challenges of being a minority. Fumitaka Matsuoka describes Asian Americans' struggles accurately when she claims that Asian Americans have to "vacillate between multiple ways of perceiving reality because they are caught between cultures and identities—subject to the dissonance of being cast in the American social imaginary as 'foreigners within' and 'model minorities.'"[13] Because Korean Americans are not treated or accepted as part of the mainstream and dominant voice in American society, they turn to Korean churches to find comfort, renew and cultivate their Korean identity, and simply to heal. Korean church communities still serve as a respite from the world. Thus, many second- and subsequent-generation KACs build their faith communities around immigrant Korean churches; consequently, their involvement in their denominations, socio-political gatherings, and community-building activities is limited. This is not to say that there is no involvement for Korean American churches in these public activities, but it simply means that because of cultural and linguistic differences, they may not be invited or have the information to participate.

---

[9]   Bhabha, *Location*, 123.
[10]  Bhabha, *Location*, pp. 147–48.
[11]  (Giffard-Foret 2013, p. 178).
[12]  Bhabha, *Location*, 169.
[13]  (Matsuoka 2011, p. 40).

## 4. Praxis of Jeong: Postcolonial Interfaith Engagement

I conducted interviews with KACs to gather stories about their relationships, engagements, and interactions that included people of other faiths. For those who participated in the interviews, the terms *interfaith dialogue* and *interfaith encounter* were foreign and new, as expected. Many do not believe they have ever engaged in interfaith dialogue or have had any interfaith encounters. However, when my research participants were asked about their interfaith engagements within a different framework, mainly that of friendships and personal relationships, they realized that they indeed have had and still have interfaith encounters. In the next section, I will investigate what these encounters look like and how they are actually interfaith engagements, albeit implicitly. Korean American Christians cultivate personal relationships in informal ways with people of other faiths and non-faiths. Some have parent groups or other types of groups in which they find people of other faiths; others live next to people of other faiths or no faith at all; and still others work with a person of a different faith. In these situations, KACs develop friendships with people of other faiths. They do not approach these relationships and friendships intentionally; rather, they happen organically. These encounters are not systemized, and Korean Americans do not need invitations to or privileged knowledge about them; they are unorganized and messy. Faith matters are neither at the forefront nor at the center of these relationships, but because people are going through life together and building trust with one another, aspects of faith flow into their relationships.

I had to unpack, however, the influence of the evangelistic aspect of the participants' religious education as far as their relationships with people of other faiths was concerned. When asked if their intentions were to share their faith in order to convert their friends and neighbors, not one said yes. They all said that they were building friendships and trusting relationships with other "human beings," not necessarily with people who had different faiths. Not much about their relationships was intentional; one exception was when participants encountered people of other faiths with food restrictions. Because this was so explicit, they had to be intentional about what to eat and when to eat, depending on the tradition.

What KACs are doing in these relationships is not a new model or a radical new way or system for people to follow. Korean Americans are simply living their lives as they know to do—organically and non-systemically building relationships with people within their reach and layering their lives with stories that include different humans with different traditions, including faith and spirituality. Interfaith interacting, for KACs, is an implicit activity—with very few intentions to encounter or to seek out people of other faiths. So, I want to make this explicit and claim that Korean Americans are actually explicitly living out jeong in these implicit interfaith relationships.

This last section will be dedicated to investigating and constructing definitions and examples of jeong to better understand it. Jeong has many non-definitions, which means that it is not only hard to define it succinctly, but also many scholars refuse to give a pithy definition. The overarching binder for jeong in KACs' interfaith relationships is human interconnectedness. Jeong is a connector that reveals to KACs that humans, despite (and because of) all of our differences, are connected to one another. Without doing too much injustice to the concept of jeong, I will point to three different categories of characteristics that will help define it and will demonstrate how KACs live out their faith through the praxis of jeong.

### 4.1. Connected by Love and Affection

An important aspect of Korean culture is han. It is a physical and visceral response to systemic oppression that all Koreans can relate to; some would claim that non-Koreans can relate to han as well.[14] What is less known about Koreans is jeong. Jeong is the other side of the same coin as han: it is

---

[14]　(Son 2014, p. 736).

an affection and love so deep and all-encompassing that it has the liberative power to heal, release, and unravel han. This powerful love and affection in jeong may be the only key to undoing han because of its inexplicable way of liberating those involved. Conversely, premature termination of jeong can result in han.[15] Both jeong and han have the potential to be stable and volatile simultaneously; the volatile side of jeong may result in obsessiveness, irrational hostility, and depression.[16] Therefore, a holistic approach that defies the obsessiveness to jeong will help humans to connect in powerful and positive ways.

Jeong is similar to compassion, but it is more than that as it leads to solidarity among people. The love and affection which jeong encompasses are not passive; it has "historically functioned as emotional bond as a survival strategy in Korea" due to the history of being attacked and annexed over time.[17] Jeong intricately weaves human strength to birth resilience in the face of trials by activating and connecting shared love and affection in humans beings. This love is fierce and does not relent. Freire describes love as something that "cannot be sentimental; as an act of freedom, it must not serve as a pretext for manipulation."[18] Love is not merely a sentiment. Freire would push the boundary for love even further by saying that if it does not "generate other acts of freedom," it is not love.[19] In the same manner, jeong as love generates freedom and contagiously compels others to do the same, creating space for solidarity. While love and affection connote gentleness and kindness, just like the han–jeong construct, love and affection have a flipside that connotes righteous indignation. However, not all have the level of courage to fight against injustice and fight for love and justice; this is when jeong's resilience and solidarity can be activated, allowing humans to cling to and strengthen one another.

The praxis of jeong in KACs' engagement in interfaith relationships is that through their personal relationships, KACs share love and affection that generate mutual warmth. No jeong is one-sided.[20] It is not like love, which can be one-directional; jeong is and must be felt by all parties involved mutually. There is a sense of interdependence in jeong through mutual love and affection. "Human" in Korean is 인간 (*in-gan: person-between*) and the Chinese character for *in* is two sticks leaning on each other (人), suggesting "that what makes us human are the relationships between us."[21] For KACs whose church teachings direct them to personalize their faith and practice it in the personal realm, this understanding of mutual love and affection for other human beings in personal relationships concretizes their faith. Given their theological tendencies toward centering the gospel in their lives and emphasizing personal convictions and conversions, envisioning practicing their faith outside of the personal relationships may seem difficult for KACs and will require unlearning (and relearning). Some of the research participants were already beginning to question confining their faith to the personal realm. As one participant said:

> Jesus is love, that Jesus loves you, is great, but that's not really laying a foundation. Like, if you have a kid who's being abused by his mom and you're thinking, "Just tell yourself Jesus loves you," what does that mean?

She, in her late twenty's, was not satisfied with simply repeating, "Jesus loves me" or "Jesus loves you" to those suffering and struggling. She wanted to see concretely how this love works and what it does to liberate. The concern she had was not self-serving, but as she paid attention to the world around her, she found herself increasingly growing uncomfortable with just "talking" about love because love and affection were more than just talk. The praxis of jeong through affection and love

---

15　(Oh 2000, p. 52).
16　Oh, *Dimensions*, 4.
17　(Choi 2010, p. 55).
18　(Freire 2011, p. 90).
19　Freire, *Pedagogy*, 90.
20　Oh, *Dimensions*, 66.
21　(Kim-Cragg 2018).

liberates and activates everyone involved. It may begin as personal curiosity, but the impact it has on human relations goes beyond our personal world.

*4.2. Connected by the Liberating and Healing Power of Jeong*

What, then, does activated love and affection look like? What is the power behind jeong? Korean scholars not only claim but also live the realities of jeong's power—power to heal, power to liberate, power to overcome oppression, and of course, power to unravel han. A Korean expression, 한이맺혀 (*han-i-met-chuh;* han is forming or knotted), is used commonly in a colloquial sense to refer to a response or emotion that stems from oppression or wrongful events. In order to ease, untangle, or unravel han, it would take something equally or more powerful. The unraveling of han requires either justice being served or healing of the wound caused by the oppression or wrongdoing. The liberative and healing traits of jeong come from the strengthening of the bond between people to overcome and move beyond the oppression as the response to it dissipates and loosens, hence unraveling the knot that exists inside a person.

For people of faith, the liberative and healing power of jeong can be found in their relationship with God. In a congregational study, Kyoo Hoon Oh writes, "it may be desirable to theologically describe our relationship of love with God as the relationship of chong. Who else besides God has ever covered our sins and understood us in such a marvelous way for such a long time?"[22] Korean Christians look to the cross and find God's powerful and liberative love that is lived out in their interpersonal relationships. In her seminal work on jeong, *The Heart of the Cross*, feminist postcolonial theologian Anne Joh treats the cross as a place where han and jeong coexist, thereby creating a space to embrace conflicting emotions and logic.[23] Joh refuses to define jeong, but the closest definition she offers is that

> *Jeong* connotes agape, eros, and filial love with compassion, empathy, solidarity, and understanding that emerges between hearts of connectedness in relationality. *Jeong* is a supplement that comes into the interstitial site of relationalism. *Jeong* is rooted in relationalism. As it emerges in between connectedness, it works as a lubricant and as relentless faith that *han* does not have the final word.[24]

The power of jeong is in relationships, as the cross is relational. In Oh's research, some Korean Christians recognize this in their practice of faith as they find "interpersonal intimacy ... more real, influential, and precious than faith in God or the Holy Spirit. In other words, the feelings and experiences of concerning a relationship with God are seen from the perspective of the growth and maturity of the interpersonal relationship."[25] It is in these intimate interpersonal relationships where the power of jeong is found; in these relationships, their han begins to unravel and jeong begins to melt the hardened hearts that have carried knotted han for so long. Furthermore, it is in these relationships that they begin to see the interdependence and interconnectedness of humanity. Despite the theological teachings and spiritual formation of conservative Korean churches, in relationships with people, whether they are Christian or not, KACs move beyond religious and faith lines and experience the power that liberates and heals. Interfaith engagement through jeong allows KACs to remain authentic to their Christian formation (being devoted to "pure" church matters and keeping their faith personal) and develop meaningful and lasting relationships and connections with people whose faith and spiritual views may be different.

---

[22] Oh, *Dimensions*, 126. Some spell jeong as chông. I have also seen jung and cheong. They are all referring to 정, jeong.
[23] (Joh 2006).
[24] Joh, *Heart*, 120.
[25] Oh, *Dimensions*, 154.

### 4.3. Connected by Stickiness and Vulnerability

Jeong creates stickiness that warmly embraces vulnerability and allows KACs to build intimate and authentic interpersonal relationships. Jeong thrives in the messiness of human relationships and seeps into the very fundamental core of our humanness. In this section, I will address how jeong's stickiness blurs boundaries in jeongful[26] relationships and how vulnerability is an essential part of developing jeong's stickiness. Then, the process through which jeong is developed will be interrogated and analyzed, as jeong requires time and space. Finally, in this section, I will demonstrate that the praxis of jeong is a pathway to overcome and correct the colonist's narrative of religious identity and theology which most Korean Christians have internalized.

In a positive light, stickiness connotes a strong bond, but negatively, it is messy. The image that I conjure up when I hear "jeong is sticky" is Elmer's glue. As a child, I used to love playing with glue; it is white but dries clear; I can put papers together, make a colorful mosaic of construction paper, and watch my craft materials stick together. What I do not like about Elmer's glue now as an adult is that it is messy. It takes a bit of adeptness to control the flow of the glue and if I press too hard, I get a glob on the paper and will have to clean it up. Sometimes, I have to wait for it to dry. Perhaps this is why adults prefer glue sticks as opposed to the liquid kind. Jeong is like Elmer's glue.

I mentioned that there is no one-sided jeong. Jeong is a mutual emotion that develops over time between two (sometimes more than two) parties. The mutual experience of jeong is because it works like glue and unites the two parties into one. It is almost that the people involved in the relationship are synthesized and "share the space of 'we.'"[27] *Woori* is we or our(s) in Korean and the sentiment behind woori is far more complex than the simple "we". Expressing possession in Korean is often done through the collective our even if it refers to the singular, my. A practical theologian and educator, Christine Hong, remembers hearing at church, "woori kyo-hwe (our church), woori ah-e-duhl (our children), woori seng-myung (our life), and even woori nah-rah (our nation)."[28] This sentiment is culturally embedded and the idea of shared space, time, possession, and even the emotions of both joy and grief permeate throughout Korean communities. Especially in Korean immigrant faith communities, "we see one another as extensions of ourselves and understand that God also sees us as extensions of one another. How one person behaves, lives, and embodies faith . . . reflects on that person's family and even larger community."[29] The already existing interconnected sentiment that Koreans innately understand joins the theological concept of each person being an extension of another, contributing to the oneness and unity felt in jeong. This unity or oneness does not mean that each individual loses his or her unique qualities; the power of jeong is that without compromising each person's unique qualities, it blurs the boundary between "I" and "you." Some scholars may even claim that mature stages of jeong lack boundaries between two parties or that in deep jeong "the individual differences are removed."[30] The removal of the difference or boundaries suggests that the two become a new entity or that the individual characteristics are not honored. Jeong's stickiness does not change the characteristics of individuals to become one, but like glue sticks pieces of paper together, it transcends these differences to cultivate and strengthen the mutual love for one another. In other words, as jeong develops and deepens, the individuals are transformed to share the collective sentiment of woori without losing their individual selves.

Jeong develops over time and frequent contact. In any form of jeong in Korea, between people, between people and place, between people and concept, between people and animals, time and

---

26 Jeongful is an adjective form of jeong which Korean American scholars use to describe relationships that have established a sturdy enough foundation to notice the presence of jeong. Oh Kyoo Hoon uses the term to describe human to human relationships as well as human to God relationships.
27 Oh, *Dimensions*, 58.
28 (Hong 2018, p. 2).
29 Hong, "Woori," 3.
30 Son, "*Jeong*," 736.

repetition nurture the growth and development of jeong. Oh uses Young-Yong Kim's definition of jeong to describe how it deepens, "[jeong] is a mental sense of ties that is unwittingly shaped through direct and/or indirect contact with and through common experiences of the given person [for a long time]"[31] In this direct and indirect contact through common experiences or shared interests, jeong develops and matures; but I add to that recipe an imperative agent, vulnerability. If a relationship develops over time and frequency of contact without vulnerability, the relationship remains superficial. Vulnerability marks trust in the person with whom the relationship is built and cultivated, and having trust points to the presence of significant history and openness. Jeongful interpersonal relationships almost demand one to "expose oneself to another with many concerns about the other's interest, character, capability, and the like, and also request that the other person expose him/herself."[32] It is vulnerability that "creatively holds together equality and difference, common sharing, and the gift of distinctiveness, and opens out into a relationality of interdependence."[33] Therefore, in the exchange of vulnerability, humans confirm the interconnectedness and further develop interdependence in their relationships. It is the stickiness that develops over time as jeong deepens and vulnerability provides the strength in the stickiness to adhere to one another in jeongful relationships.

Jeong's development over time and space deserves an in-depth look. Through frequency and usually over a long period of time, jeong develops or, as I like to say, seeps into those involved. It is not an intentional process or engineered formula for jeong to develop and mature, but as the word seep implies, it naturally soaks through its surroundings and penetrates into the lives of those in the relationship. It is organic. It is chaotic, with no systems involved. It usually develops over a long period of time because relationships take time with shared history, contact, and trust. Once jeong begins to seep into a relationship, it begins to defy temporal and spatial realities. Emily Lee, a philosophy scholar, uses Maurice Merleau-Ponty's phenomenological framework to bring attention to the postcolonial concept of past, present, and future.[34] Treating the concept of past, present, and future, Merleau-Ponty focuses on the act of "recalling" and shows that "one only remembers the past through the lens of the present."[35] What holds past, present, and future together is the present and the motivations for recalling at the time of recalling. Lee writes, "Merleau-Ponty posits that even though the absolute past and present do not overlap, moments of time do overlap."[36] To better understand this concept of overlapping, Merleau-Ponty explains that the absolute event in the past (B) when recalled in the present moment (C), not only leads to the present moment (C), but also to a different version of the past (B'). Therefore, in the present moment, both the recalled versions of the past exist in the same time and space. Jeong defies time in the similar sense. The sense of stickiness and messiness of jeong in the present moment must be understood not only as a product of the past, history, vulnerability, trust, love and affection, but also as having those components coexist in the same space and time. In other words, jeong develops, matures, and sustains holistically, not bound by time.

Thus, the powerful nature of jeong is rather complex. In the previous section, jeong's power to heal and liberate was discussed. In that power to heal and liberate is an opportunity for KACs to correct the imposed colonial religious identity and regenerate and re-narrate their own story situated in the unique Korean American context. Korean Christians have a negative, even inferior, view of their own traditional culture because of "the influences of the traditional theology influenced by American missionaries."[37] Some might even claim that Korean culture, with shamanistic tendencies, is evil. What Korean Christians do not realize is that once the colonial narratives are exposed and they begin to understand how their Christian formation was used to keep them in neat compartments which are easy

---

31 Oh, *Dimensions*, 46.
32 Oh, *Dimensions*, 58.
33 (Reynolds 2012, p. 221).
34 (Lee 2008, p. 549).
35 Lee, "Phenomenology," 549.
36 Lee, "Phenomenology," 549.
37 Oh, *Dimensions*, 39.

to control, they may recognize a ball of han raising its head inside them. The injustice of erasing and usurping one's cultural expressions in the name of Christianity may result in han. However, Joh claims that "rising out of the connectedness of hearts, *jeong* emerges in a transformative becoming within the interstitial space between the self and the other, a becoming that transcends *han*."[38] Jeong creates the very needed space and opportunity without threatening KACs in their understanding of faith practices to transcend the rhetoric and narrative they have heard from the colonists and American evangelical theologians to engage in interfaith relationships and encounters. While some engagement may be intentional, most of these relationships arise out of being in the same workplace, or the same neighborhood, going to the same gym, coffeeshop, or farmer's market, or having children or pets that are friends. From these organic and (un)systemized relationships will come cognitive understanding and curiosity as they become more aware of jeong developing in them.

## 5. Conclusions

　　Korean American Christians with or without awareness have been operating under the dominant Western evangelical Christian narrative and identity. Korean Christian history coupled with deep-seated Confucian teachings led Korean Christians to receive Western Christianity without much questioning or critical thinking. I agree with some who claim that Korean Christians perceive their cultural expressions of spirituality as inferior or even evil. Inevitably, Korean Christian spiritual formation rejected some Korean cultural expressions of spirituality. However, jeong, an emotional response and bond that all Koreans and Korean Americans understand and build, continues to seep into Korean and Korean American Christians' lives to move beyond their cognitive understanding of the Western evangelical Christian narrative that was imposed on them. Furthermore, Korean American Christians participate in interfaith encounters in an organic, non-systemized, and messy way through the praxis of jeong. The next steps for Korean American Christians in interfaith engagement may be to make their implicit and muted interfaith relationships more explicit by becoming aware of their own Christian history and reflecting on their Christian identity.

**Funding:** This research received no external funding.

**Conflicts of Interest:** The author declares no conflict of interest.

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
