# Peer review of "Jeong: A Practical Theology of Postcolonial Interfaith Relations"

_religions, doi:10.3390/rel11100515_

Round 1

Reviewer 1 Report

I want to commend the author for offering a flawless essay--this is beautifully written, masterfully organized, expertly researched, and offers a compelling argument about the perseverance of Korean cultural norms within changing geopolitical and religious landscapes.  There's nothing really to criticize, its great work.

Author Response

Thank you so much. 

Reviewer 2 Report

SPECIFIC COMMENTS

Line 85            “I interviewed…”

            I expect more detailed information about the research design in “Religions”

– N=??

– When, Where, How…

Line 94            “criteria”

– That is a plural, so the verb should agree. Or it should read “The basic criterion…. is:”

But, actually, it is not the right word. The quotation offers a DEFINITION not a criterion.

So:       “The basic definition of evangelical, even though it is a broad term…”

Line 243          “Some of the research participants…”

            Again, I am frustrated, not knowing age, gender, location, etc.

            The quotation at lines 245-6-7 – is she young, a university student? Older? Married?

            Was the interview in English? Was translation part of your research path?

Line 294 and earlier and later            “stickiness”

            I get it. And I think the article can be published as-is.

            But the author might consult a dictionary of the social sciences, and read a few paragraphs on “social cohesion,” This has been a major theme in the study of religion and society since Durkheim and Weber, since Marx.

            The functionalist approach to the social scientific study of religion emphasizes the creation of relationship, of community, of social group… social cohesion.

Line 353 and elsewhere                      “seeps into”

            Again, I get it.

            But use of this vocabulary fails to connect the scholarship with all the literature on socialization and the internalization of values. Again, the functionalist would emphasize that religion provides a sense of identity… for the individual and the group.

BOTTOM LINE

This is a very worthwhile article, and I recommend that it be published.

MORE GENERAL COMMENTS

The author might be interested in my reaction below:

Thank you for the well-written article on “Jeong” and KACs.

If you have the time, or if you pursue this topic, you may be interested in this response from my unique perspective.

I am a Canadian, an ordained minister in The United Church of Canada with a PhD in Sociology. I have had a 45-year career in church leadership and ecumenical and interfaith activity, and a parallel part-time career for 35 years teaching Anthropology, Sociology and Religion courses at two Canadian universities (graduate and undergraduate courses, and thesis supervision).

Among decades of ecumenical and interfaith engagements, for example, I was part of the United Church delegation to the Tenth Assembly of the World Council of Churches in Busan Korea in 2013.

I recognize the phenomenon you discuss in the paper, but I would have appreciated a clearer statement that your discussion is limited to Korean Christians in the United States, and that it cannot be generalized beyond that society.

You make no mention of particular denominations and their distinct cultures.

Korean Christians in Canada behave in several different ways.

In the United Church, they tend to be justice-concerned (“social gospel”), progressive (or “liberal” as some say), and engaged with other groups in society, other congregations, other denominations, other religious groups, other groups that may be non-religious but share an agenda of justice, refugee sponsorship, environmental concern, anti-racism, etc. There are Korean congregations, but Christians of Korean descent who prefer the United Church are often part of the diversity of their local congregation.

Five of the 10 largest congregations in the Presbyterian Church in Canada are Korean-speaking. Korean Canadian Presbyterians tend to participate less actively on that list of engagements for United Church Korean Christians (above), but they are present and active.

The United Church of Canada elected the Rev. Sang Chul Lee as Moderator in 1988.  The Presbyterian Church in Canada elected the Rev. Cheol Soon Park as Moderator in 2008.

Other Canadian Christians of Korean background may prefer to identify themselves by using the words of the Shah-Forster definition (not criterion) referred to above, “A Protestant who…decision.” (lines 97-100.

Those people will be found in “Alliance” congregations (Christian Missionary Alliance) and various independent congregations that identify as “Bible churches” or perhaps “Brethren.”

In my experience as a university chaplain in Canada for part of my ministry career, I found that Korean students who wished to participate in the Body of Christ made various decisions. Some sought out a Korean congregation of their favourite denomination, if it was available. Some chose a neighbourhood congregation for specific personal reasons (proximity, time of services, music program, preaching, friends and relatives, etc.) Some stayed on campus and EITHER integrated themselves into the diverse chaplaincy programs OR participated in a Korean Christian group, usually called something like “Bible Study Club.”

The issue of cultivating “their Korean identity” (line 160) is different in Canada and the United States. Canada has had more than 40 years of intentional, government-funded multicultural programming to encourage respect for and stewardship of cultural heritages.

Racism and multiculturalism are very different in Canada and the United States. A Canadian of Korean origin may celebrate that identity, listen to Korean media, read Korean periodicals, speak Korean, etc.  An American of Korean origin is part of the BAPOC population, struggling for constitutional rights and personal safety. That’s quite a distinction.

And from my international, ecumenical and interfaith networks, I know that statements about KACs do not necessarily apply in Britain, Australia or New Zealand either.

Please clarify that, if you would.

These have been the comments of a United Church minister and Sociologist.

As a theologian, and reader of Religions, I have another request for you to consider, at least for the future.

I wish you had discussed Jeong and Han in terms of ἀγάπη and ἁμαρτία – agapeic love and sin.

The section beginning at line 232 could have been related to the early Jesus communities of sharing and mutual support such as we find in Acts chapter 2 and chapter 4, and referred to in some of the epistles.

Author Response

Thank you for your comments. I made changes reflecting your suggestions.

When the time comes and if possible, I'd love a conversation with you to hear from your unique contexts and perspectives.

Reviewer 3 Report

This article offers an original take on interfaith relations by adopting an indigenous concept as a lens for examining such relations within an ethnic minority practicing the dominant religion of the host culture. In addition, it gives attention to the manner in which Korean Americans conceive of interreligious relations, a matter that has not been given scholarly attention that I am aware of. It will be a welcome contribution to the emerging field of interfaith studies. I attach include in this review a pdf with a few editorial changes that I think are necessary to correct some awkward or colloquial phrasing. Those sentences aside, the piece is polished and carefully written.

While the article could be improved by a more robust engagement with scholarly literature, including literature from interreligious studies, I find the author's engagement with theory to be more than adequate and itself a contribution to interreligious studies, which to this point has not examined deeply its connection to theory. I recommend publication following the editorial corrections I suggest. 

Author Response

Thank you for your comments. I made changes to reflect your suggestions.